# Show-through removal with sparsity-based blind deconvolution

**Sota Kawakami**[1]\*, **Hiroyuki Kudo**[2]

**1** Degree Programs in Systems and Information Engineering, Graduate School of Science and Technology, University of Tsukuba, Tsukuba, Japan, **2** Institute of Systems and Information Engineering, University of Tsukuba, Tsukuba, Japan

\* kawakami.sota21@gmail.com

## Abstract

When scanning a document printed on both sides by using an electronic scanner, the printed material on the back (front) side may be transmitted to the front (back) side. This phenomenon is called show-through. The problem to remove the show-through from scanned images is called the show-through removal in the literature. In this paper, we propose a new method of show-through removal based on the following principle. The proposed method uses two scanned images with the front side and with the back side as input images. The proposed method is based on Ahmed's Blind Image Deconvolution method discovered in 2013, which succeeded in formulating Blind Image Deconvolution as a nuclear norm minimization. Since the structure of show-through removal resembles that of Blind Image Deconvolution, we discovered that the show-through removal can be reformulated into a nuclear norm minimization in the space of outer product matrix constructed from an image vector and a point spread function vector of blurring. Using this key idea, we constructed the proposed method as follows. First, our cost function consists of the following three terms. The first term is the data term and the second term is the nuclear norm derived from the above reformulation. The third term is a regularization term to overcome the underdetermined nature of show-through removal problem and the existence of noise in the measured images. The regularization term consists of Total Variation imposed on the images. The resulting nuclear norm minimization problem is solved by using Accelerated Proximal Gradient method and Singular Value Projection with some problem-specific modifications, which converges fast and requires a simple implementation. We show results of simulation studies as well as results of real image experiments to demonstrate the performances of the proposed method.

## 1 Introduction

In recent years, digitization of paper documents has become very popular. When the paper documents are digitized, a phenomenon called show-through may occur, in which the printed material on the back (front) side is transmitted to the front (back) side. It reduces visibility of text and accuracy of OCR (Optical Character Recognition). The problem to remove the show-

**Data Availability Statement:** All code and experiment image files are available from the Zenodo repository (https://zenodo.org/records/11016261).

**Funding:** The author(s) received no specific funding for this work.

**Competing interests:** The authors have declared that no competing interests exist.

through from scanned images is called the show-through removal in the literature. Up to now, several methods of the show-through removal have been proposed. For example, the method using binarization [1, 2] and the method using color tone correction [3] have been used in electronic scanners. This is because the process of show-through removal in these methods is simple and requires only a single input image. However, these methods tend to be inaccurate for the printed matter having gradation such as images and drawings. On the other hand, other improved methods for the show-through removal have been proposed, in which both the scanned paper of front page and that of back page are used as input images. There exist a method using Kalman filter [4, 5], a method using Independent Component Analysis (ICA) [6–8], a method using Non-negative Matrix Factorization (NMF) [9], a method using Total Variation (TV) [10], and a method using adaptive filtering [11, 12]. These methods achieve higher accuracy compared to the earlier methods. However, each of these methods still suffers from drawbacks such as difficulty in adjusting parameters, inability to process well when there exists a printed matter on both the front and back pages, and dependency of the solution on initial values of iterative methods.

Also, in recent years, show-through removal methods using Convolutional Neural Network (CNN) have been proposed [13, 14]. Among them, the method based on Auto-encoder [14] has a large potential. A typical major application of Auto-encoder is to remove disturbance from signals or images, which has been already proposed in image denoising, speckle reduction in SAR or ultrasound images, and clutter rejection in radar images. The same method can be also used to remove show-through from scanned images. This method requires only a single image of front side. However, its accuracy depends on used training data, and it may be weaker to the show-through having strong correlation with the front page image. On the other hand, our proposed method requires both of an image of the fore side and that of the back side, but using the two images eliminates the necessity of training data and may lead to an improved accuracy (for example, in the above mentioned case). At the current stage, we believe that both the optimization-based approach like ours and the Auto-encoder-based approach are meaningful to study, because they have different advantages and disadvantages and the both may be able to be combined in the future to develop more accurate or more convenient methods.

In this paper, we propose a new method of show-through removal based on the nuclear norm minimization. The principle of this method is summarized as follows. First, the proposed method uses two scanned images with the front side and with the back side as input images. The proposed method is based on Ahmed's Blind Image Deconvolution (BID) method [15] discovered in 2013. Most classical methods for BID were based on a non-convex optimization, but Ahmed *et al.* succeeded in formulating BID as a nuclear norm minimization, which is one of famous convex optimization problems appearing in Compressed Sensing (CS). The key of their work lies in reformulating BID in the space of outer product matrix constructed from an image vector and a Point Spread Function (P.S.F.) vector of blurring. Since the structure of show-through removal resembles that of BID, we discovered that the show-through removal can be reformulated into a nuclear norm minimization in the space of outer product matrix constructed from an image vector and a P.S.F. vector. We have succeeded in this reformulation by following the similar derivation step to Ahmed's BID method. Using this key idea, we constructed the proposed method as follows. First, our cost function for minimization consists of the following three terms. The first term is the data term and the second term is the nuclear norm derived from the above reformulation. However, without a regularization term, the method does not work well due to the underdetermined nature of show-through removal problem and the existence of noise in the input images. Therefore, we introduce a regularization term, which is TV imposed on the images. The resulting nuclear minimization problem is solved by using Accelerated Proximal Gradient (APG) method [16] and

Singular Value Projection (SVP) [17] with some problem-specific modifications, which converges fast and requires a simple implementation. In addition, the implementation of APG method is very simple.

The paper is organized as follows. In Section 2, we describe a mathematical model of show-through removal. In Section 3, we briefly review Ahmed's BID method which is the basis of this work. In Section 4, we explain the proposed method in detail. The explanation mainly consists of 1) the derivation to convert the show-through removal into a nuclear norm minimization and 2) the algorithm description of the proposed method. In Section 5, we show results of simulation studies as well as results of real image experiments. The conclusion is described in Section 6.

## 2 Mathematical model of show-through removal

Before explaining the proposed method for show-through removal, we explain a mathematical model of image degradation by the show-through. Due to the show-through, pixel values of the original image on the front side are affected by the blurred image of the back side image and the transmittance (from the back side to the front side). The blurring is caused due to diffusion of light occurring when the light passes through the paper. Using equations, the standard model of show-through can be expressed as

$$
\begin{aligned}
G_1 &= T(F_1, H, M(F_2), a) \\
G_2 &= T(F_2, H, M(F_1), a)
\end{aligned}, \tag{1}
$$

where $G_1$ is the image of front side with show-through, $G_2$ is the image of back side with show-through, $F_1$ is the original image of front side, $F_2$ is the original image of back side. Furthermore, $H$ is the P.S.F. of the above-mentioned blurring and the parameter $a$ is the transmittance. Furthermore, $M(\cdot)$ is the operator which flips the image horizontally and $T(\cdot)$ is the linear operator which represents the degradation caused by the show-through. We note that this linear degradation model of show-through was described in [5].

Next, we describe the detailed form of operator $T(\cdot)$ appearing in Eq (1). The operator $T(\cdot)$ is expressed as

$$
\begin{aligned}
B_1(x,y) &= a \sum_{y'=\frac{1-K_y}{2}}^{\frac{K_y-1}{2}} \sum_{x'=\frac{1-K_x}{2}}^{\frac{K_x-1}{2}} (L_{\max} - M(F_1)(x+x', y+y'))H(x',y') \\
B_2(x,y) &= a \sum_{y'=\frac{1-K_y}{2}}^{\frac{K_y-1}{2}} \sum_{x'=\frac{1-K_x}{2}}^{\frac{K_x-1}{2}} (L_{\max} - M(F_2)(x+x', y+y'))H(x',y') \\
G_1(x,y) &= F_1(x,y) - B_2(x,y) \\
G_2(x,y) &= F_2(x,y) - B_1(x,y)
\end{aligned}. \tag{2}
$$

In Eq (2), $B_1(x,y)$ is the horizontally flipped blurred image of front side with inverted pixel value and gain $a$, $B_2(x,y)$ is the horizontally flipped blurred image of back side with inverted pixel value and gain $a$. $L_{\max}$ is the maximum value of the image (255 for 8-bit images), and $K_x$ and $K_y$ are width and height of P.S.F. $H$, respectively. In addition, $H(0,0)$ is

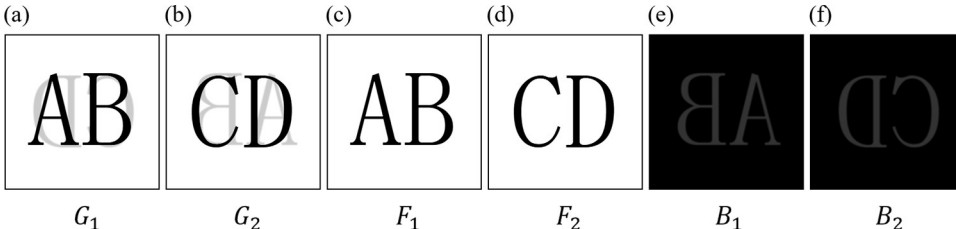

**Fig 1. Example images constructed by using the show-through model.** (a) $G_1$ degraded image (front side), (b) $G_2$ degraded image (back side), (c) $F_1$ original image (front side), (d) $F_2$ original image (back side), (e) $B_1$ horizontally flipped blurred image with invert pixel and gain $a$ (front side), (f) $B_2$ horizontally flipped blurred image with invert pixel and gain $a$ (back side).

defined as the center of $H$, not the first element. Also, we assume that $F_1(x,y)$ and $M(F_2)(x+x',y+y')$ (as well as $F_2(x,y)$ and $M(F_1)(x+x',y+y')$) are uncorrelated to each other. In Fig 1, we show example images constructed by using this model. The show-through removal can be formulated as a problem to restore a pair of original images $F_1$ and $F_2$ from a pair of observed images $G_1$ and $G_2$ degraded by the show-through. We remark that, throughout this paper, both the scanned paper of the front side and that of the back side are available for the restoration, because using the two input images allows higher restoration accuracy as demonstrated in [4–12].

## 3 Ahmed's blind image deconvolution method

The proposed method for show-through removal is largely based on Ahmed's BID method [15] discovered in 2013. In this section, we briefly review Ahmed's BID method for unfamiliar readers. First, we explain image degradation model by the blurring. Let the original image be $F$ and the P.S.F. of blurring be $H$. Then, the blurred image $S$ can be expressed as

$$S = F \otimes H, \tag{3}$$

where the symbol $\otimes$ denotes two-dimensional (2-D) convolution. Let $S(x,y)$, $F(x,y)$, and $H(x,y)$ be values of $S$, $F$, and $H$ at pixel $(x,y)$. Then, Eq (3) is expressed as

$$S(x,y) = \sum_{y'=\frac{1-K_y}{2}}^{\frac{K_y-1}{2}} \sum_{x'=\frac{1-K_x}{2}}^{\frac{K_x-1}{2}} F(x+x',y+y')H(x',y'), \tag{4}$$
$$(x = 1, 2 \cdots M_x, y = 1, 2 \cdots M_y)$$

where $M_x$ and $M_y$ are width and height of $F$, respectively. BID is a problem of estimating the original image $F$ and the P.S.F. $H$ simultaneously from the degraded image $S$. Most conventional BID methods are based on a non-convex optimization by using an iterative method [18–20]. On the other hand, in 2013, Ahmed *et al.* [15] succeeded in formulating BID into a form of nuclear norm minimization, which is one of famous convex optimizations problems appearing in CS. The key of their work lies in reformulating BID in the space of outer product matrix constructed by the image vector and the P.S.F. vector. We briefly describe this derivation step below. Let $\overrightarrow{f} = (f_1, f_2, \cdots f_{M_x M_y})^T$ be the vectorized form of $F$. Let $\overrightarrow{h} = (h_1, h_1, \cdots h_{K_x K_y})^T$ be the vectorized form of $H$. We assume that the vectorization of image and P.S.F. is performed according to the raster-scan order. In Ahmed's BID method, an outer

product matrix $X$ constructed from $\overrightarrow{f}$ and $\overrightarrow{h}$ is used, where $X$ is defined by

$$X = \overrightarrow{f}\,\overrightarrow{h}^{T} = \begin{pmatrix} f_1 h_1 & f_1 h_2 & \cdots & f_1 h_{K_x K_y} \\ f_2 h_1 & f_2 h_2 & \cdots & f_2 h_{K_x K_y} \\ \vdots & \vdots & \ddots & \vdots \\ f_{M_x M_y} h_1 & f_{M_x M_y} h_2 & \cdots & f_{M_x M_y} h_{K_x K_y} \end{pmatrix}. \tag{5}$$

By the definition of outer product, the rank of matrix $X$ is always 1 when the two vectors $\overrightarrow{f}$ and $\overrightarrow{h}$ are non-zero. Then, by using $X$, we can perform the same calculation as Eq (4) representing the convolution by

$$S(x,y) = \sum_{y'=\frac{1-K_y}{2}}^{\frac{K_y-1}{2}} \sum_{x'=\frac{1-K_x}{2}}^{\frac{K_x-1}{2}} X\left( (y' + \frac{K_y-1}{2})K_x + x' + \frac{K_x+1}{2}, (y+y')M_x + x + x' \right), \tag{6}$$

where the convolution could be expressed with a simple linear operator which acts on the matrix $X$. We denote this linear operator by $\mathcal{A}(\cdot)$. Then, Eq (6) is expressed as

$$\overrightarrow{s} = \mathcal{A}(X), \tag{7}$$

where $\overrightarrow{s}$ is a vector defined by arranging all elements of image $S$ according to the raster-scan order. Therefore, BID can be formulated as a problem to find a matrix $X$ satisfying Eq (7) and having a rank of 1. This problem is expressed as

$$\min_X \text{rank}(X) \text{ subject to } \overrightarrow{s} = \mathcal{A}(X). \tag{8}$$

The problem of finding a matrix with rank of 1 is called the rank minimization in the literature. It is known that computational complexity of the rank minimization is NP-hard, but it can be solved with sufficient accuracy in practice by using a convex relaxation into a nuclear norm minimization. This fact has been demonstrated in a variety of problems in real world [21]. The nuclear norm minimization corresponding to Eq (8) is expressed as

$$\min_X \|X\|_* \text{ subject to } \overrightarrow{s} = \mathcal{A}(X). \tag{9}$$

Through the above derivation step, Ahmed *et al.* succeeded in formulating BID by a convex optimization. We remark again that the conventional methods of BID [18–20] are based on a non-convex optimization so that the solution depends on initial values of image and P.S.F. used in the iterative method. Therefore, we expect that Ahmed's BID method allows more stable and more accurate restoration.

## 4 Proposed method for show-through removal

### 4.1 Application of Ahmed's BID method to show-through removal

Let us consider the problem of show-through removal formulated in Section 2. The show-through removal method in this paper estimates the two original images $F_1$ and $F_2$ from the two input degraded images $G_1$ and $G_2$. First, in order to extend Ahmed's BID method to the show-through removal, the definition of outer product $X$ and the degradation model need to be modified as follows. We note that this modification is a key in this work. We define the

outer product matrix $X$ constructed from $F_1, F_2, H, L_{\max}$ and $a$ by

$$X = \overrightarrow{f}\,\overrightarrow{h}^T$$
$$\overrightarrow{f} = (L_{\max} - f_{1_1}, L_{\max} - f_{1_2}, \cdots, L_{\max} - f_{1_{M_x M_y}}, L_{\max} - f_{2_1}, L_{\max} - f_{2_2}, \cdots, L_{\max} - f_{2_{M_x M_y}})^{\mathrm{T}}, \quad (10)$$
$$\overrightarrow{h} = (ah_1, ah_2, \cdots, ah_{K_x K_y}, 1)^{\mathrm{T}}$$

where $\overrightarrow{f}$ is a vector defined by arranging all elements of images $F_1$ and $F_2$ (subtracted from $L_{\max}$) according to the raster-scan order. This vector takes the inversion of pixel values appearing in the show-through model of Eq (2) into account. Also, in the P.S.F. vector $\overrightarrow{h}$ defined by arranging all elements of P.S.F. $H$ according to the raster-scan order, the first $K_x K_y$ elements of $\overrightarrow{h}$ include the multiplication by the transmittance $a$. This definition allows us to embed both the degradation due to blur $H$ and the effect of transmittance $a$ simultaneously in $\overrightarrow{h}$. As a result, we no longer need to estimate the P.S.F. and the transmittance separately. Furthermore, $K_x K_y$+1-th (last) element of $\overrightarrow{h}$ constructs $K_x K_y$+1-th column of matrix $X$ which is equivalent to the image $\overrightarrow{f}$.

By defining the matrix $X$ in this way, similarly to Eq (6), the degraded images $G_1$ and $G_2$ defined in Eq (2) can be calculated from the matrix $X$ by

$$F_1(x,y) = L_{\max} - X(K_x K_y + 1, yM_x + x)$$
$$F_2(x,y) = L_{\max} - X(K_x K_y + 1, M_x M_y + yM_x + x)$$
$$B_1(x,y) = \sum_{y' = \frac{1 - K_y}{2}}^{\frac{K_y - 1}{2}} \sum_{x' = \frac{1 - K_x}{2}}^{\frac{K_x - 1}{2}} X\left(\left(y' + \frac{K_y - 1}{2}\right)K_x + x' + \frac{K_x + 1}{2}, (y + y')M_x + x + x'\right)$$
$$B_2(x,y) = \sum_{y' = \frac{1 - K_y}{2}}^{\frac{K_y - 1}{2}} \sum_{x' = \frac{1 - K_x}{2}}^{\frac{K_x - 1}{2}} X\left(\left(y' + \frac{K_y - 1}{2}\right)K_x + x' + \frac{K_x + 1}{2}, M_x M_y + (y + y')M_x + x + x'\right)$$
$$.(11)$$
$$G_1(x,y) = F_1(x,y) - M(B_2)(x,y)$$
$$G_2(x,y) = F_2(x,y) - M(B_1)(x,y)$$
$$(x = 1, 2 \cdots M_x, y = 1, 2 \cdots M_y)$$

In Fig 2, we show the structure of matrix $X$ and the process of generating $G_1$ and $G_2$ from the matrix $X$ by Eq (11).

Furthermore, if we define the blurred image vector $\overrightarrow{g}$ by linking the vectorized forms of $G_1$ and $G_2$, Eq (11) can be expressed by using the linear operator $\mathcal{T}(\cdot)$ acting on the matrix $X$ as

$$\overrightarrow{g} = \mathcal{T}(X)$$
$$\overrightarrow{g} = (g_{1_1}, g_{1_2}, \cdots, g_{1_{M_x M_y}}, g_{2_1}, g_{2_2}, \cdots, g_{2_{M_x M_y}})^{\mathrm{T}} . \quad (12)$$

According to the above derivation, we succeeded in modelling the image degradation by show-through using the outer product matrix $X$. Furthermore, from the definition of outer product, we know that the rank of matrix $X$ is always 1 when the two vectors $\overrightarrow{f}$ and $\overrightarrow{h}$ are non-zero. So, by applying the same nuclear norm minimization as in Ahmed's BID method, $i.$ $e.$ Eq (9) in the BID case, the show-through removal can be formulated as the following

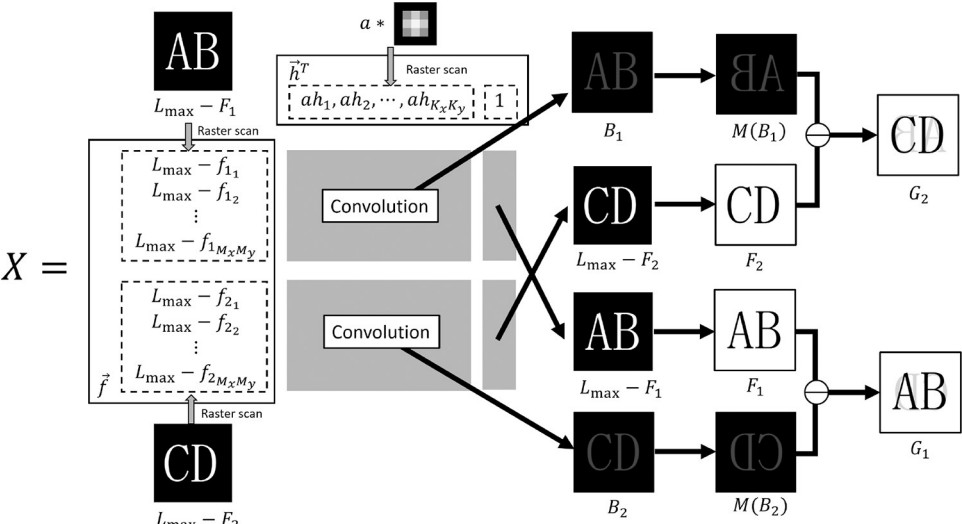

**Fig 2. Schematic diagram of the structure of matrix $X$ and the generation process of $G_1$ and $G_2$ from matrix $X$.**

optimization problem.

$$\min_X \|X\|_* \text{ subject to } \overrightarrow{g} = \mathcal{T}(X) \qquad (13)$$

## 4.2 Description of proposed algorithm

There exist mainly the following two drawbacks in original Ahmed's BID method. The first drawback is lack of regularization term. Without the regularization term, it does not work well when the problem itself is underdetermined or when the degraded images contain noise. The second drawback is large computational cost, because Ahmed *et al.* used a slow and inefficient optimization method to solve Eq (9), *i.e.* general solver of semi-definite programming. To overcome these drawbacks, we have used the similar approach to that in our previous work on the standard BID and motionless super-resolution [22]. For the regularization, we have used a regularization term consisting of TV penalty imposed on the images. To reduce the computational cost, we have used APG method [16] and SVP [17] with some problem-specific modifications to solve Eq (13). This section describes the proposed algorithm in detail.

**4.2.1 Regularization term.** In its original form, Ahmed's BID method used some unrealistic constraint such as *a priori* knowledge on the object support or a distribution of zero coefficient in some transformed (sparsified) space to overcome the problem of undetermined nature of BID problem, because the number of estimated variables is larger than the number of measured data in this problem. To overcome this drawback, we have proposed the use of regularization term [22]. Our regularization term consists of the TV penalty imposed on the images $F_1$ and $F_2$ to be estimated. Thanks to the regularization term described below, we expect that accuracy and stability of show-through removal can be improved significantly.

We denote the TV penalty imposed on the images $F_1$ and $F_2$ by $R(X)$. It is well-known that TV possesses an effect of smoothing flat parts of the image while edge parts are strongly preserved. However, we remark that, in the case of show-through removal, TV must be applied to the optimization variable $X$ (not to the image vector $\overrightarrow{f}$). Concretely, $R(X)$ applied to $X$ is

expressed as

$$\overrightarrow{y}_h(i) = \begin{cases} X(K_xK_y + 1, i + 1) - X(K_xK_y + 1, i)(\text{if } i \bmod M_x \neq 0) \\[2em] 0 \text{ (otherwise)} \end{cases}$$

$$(i = 1, 2, \cdots, 2M_xM_y)$$

$$\overrightarrow{y}_v(i) = \begin{cases} X\left(K_xK_y + 1, i + M_x\right) - X\left(K_xK_y + 1, i\right)\left(\text{if } \mathrm{ceil}\left(\dfrac{i}{M_x}\right) \neq M_y\right) , \\[2em] 0 \text{ (otherwise)} \end{cases} \quad (14)$$

$$(i = 1, 2, \cdots, 2M_xM_y)$$

$$R(X) = \sum_{i=1}^{2M_xM_y} \sqrt{\overrightarrow{y}_h(i)^2 + \overrightarrow{y}_v(i)^2 + \varepsilon}$$

where $\varepsilon > 0$ is a small positive number. The rationale behind the definition of Eq (14) is as follows. Fortunately, in the definition of matrix $X$, $K_xK_y$+1-th column of $X$ coincides with $\overrightarrow{f}$ as mentioned in Eq (10). Therefore, in Eq (14), TV calculated from $K_xK_y$+1-th column of $X$ can be directly used as $R(X)$, which corresponds to applying TV on the images. We believe that this definition of $R(X)$ is a clever idea.

Finally, by including this regularization term, our regularized counterpart of Ahmed's BID method to solve Eq (13) can be formulated as the following optimization problem.

$$\min_X \alpha\|X\|_* + \beta R(X) \text{ subject to } \overrightarrow{g} = \mathcal{T}(X) \qquad (15)$$

When the noise is contained in the degraded input images, it is necessary to change the data fidelity term from the hard constraint into the least-squares cost function, leading to the following optimization problem.

$$\min_X \alpha\|X\|_* + \beta R(X) + \frac{1}{2}\|\mathcal{T}(X) - \overrightarrow{g}\|^2 \qquad (16)$$

**4.2.2 Optimization method.** In CS, the nuclear norm minimization problem is usually solved by using iterative Soft-Thresholding (ST) algorithm, which is known to be a particular application of Proximal Gradient (PG) method. Although this method is known to converge to an exact solution, it is necessary to compute all singular values of matrix $X$ at every iteration. The computation of all the singular values of a large matrix significantly increases computational cost. Furthermore, the speed of convergence is also very slow, because it takes a longer time for the effect of ST to be reflected to the solution. Therefore, we used SVP instead of ST to compute the proximal mapping of nuclear norm. Although using SVP makes it impossible to solve the nuclear norm minimization exactly, it improves practical performances of the proposed method significantly in terms of convergence speed and computational efficiency [17]. In our problem, we know that the rank of $X$ is 1 in prior to the restoration. So, we calculated only one largest singular value and discarded other singular values without computing them at each iteration. In addition, Eq (16) is solved by using APG method [16], which is known to converge faster than the standard PG method. APG method with the above SVP modification in the thresholding is summarized in ALGORITHM 1. We note that Step 4 in ALGORITHM 1 corresponds to the modified SVP part. Furthermore, in order to stabilize the restoration, elements of intermediate matrix $W^k$ being estimated as less than 0 or greater than $L_{\max}$ was

projected to the interval $[0, L_{\max}]$ in Step 3 of ALGORITHM 1. This modification also improved accuracy of the restoration in addition to the SVP modification.

```
ALGORITHM 1: ACCELATED PROXIMAL GRADIENT METHOD
```

Input : $X^0 = \vec{f}^0 \vec{h}^{0T}$

Output : $\vec{f}^k, \vec{h}^k$

Initialization : $t^0 = t^{-1} = 1.0, X^{-1} = X^0$

$k \leftarrow k+1$

$Z^k = X^k + \frac{t^{k-1}-1}{t^k}(X^k - X^{k-1})$

$W^k = Z^k - (\eta^k)^{-1}\{\beta \nabla R(Z^k) + \mathcal{T}^*(\mathcal{T}(Z^k) - \vec{g})\}$

$W^k = \min(L_{\max}, \max(0, W^k))$

$W^k \vec{U} D V^T, D = \mathrm{diag}(\sigma_1, \sigma_2, \cdots, \sigma_r)$

$\quad X^{k+1} = U \mathrm{diag}(\sigma_1, 0, \cdots, 0) V^T$

$t^{k+1} = \frac{1+\sqrt{1+4(t^k)^2}}{2}$

$\quad$ Calculate $\vec{f}^k, \vec{h}^k$ from $X^k$

## 5 Experimental studies

### 5.1 Simulation studies

First, we performed several simulation studies by using artificially generated degraded images by the show-through. All images used in simulation experiments in this paper were artificially generated by ourselves. In all the simulation studies below, the following parameter values of image degradation were used. The P.S.F. to blur the back (front) side image was a Gaussian filter with $\sigma = 1.0$ and the size of support was set to 3×3 (pixels). The value of transmittance was set to $a = 0.1$. Conditions in implementing the proposed method are summarized as follows. Initial values of $F_1$ and $F_2$ in the APG iteration were set to the degraded images $G_1$ and $G_2$. The size of estimated P.S.F. was set to 5×5 (pixels), which is larger than the true size of P.S.F., and initial values of P.S.F. in the APG iteration were set to

$$H(x,y) = \begin{cases} \dfrac{1}{25} & (-2 \leq x \leq 2, -2 \leq y \leq 2) \\ 0 & \text{(otherwise)} \end{cases}. \tag{17}$$

Initial values of transmittance was set to $a = 0.5$. The parameter values in the proposed method were set to $\beta = 2.5$ in Eq (16) and $\varepsilon = 1$ in Eq (14). Finally, the step-size value in Step 2 of ALGORITHM 1 was set to $(\eta^k)^{-1} = 0.05$, and the number of iterations was 200. The hardware and software environment used in the experimental studies is summarized in Table 1. All the images and codes used in the experiments are available from "https://zenodo.org/records/11016261". All the experiments can be reproduced.

**5.1.1 Experiment to confirm the effects of regularization term.** First, we confirmed the effects of TV regularization term described in Section 4.2.1. Image restorations were performed with and without the TV penalty. We used two images (S1 File) shown in Fig 3(A) and 3(B) for Experiment 1 (Fig 4(A) and 4(B) for Experiment 2), which simulate the printed

**Table 1. Hardware and software environments used in the experimental studies.**

| OS | Windows 10 64bit |
|---|---|
| CPU | Intel(R) Core(TM) i7-3770 CPU @ 3.40GHz |
| RAM | 32.0 GB |
| Programming Language | C++ |
| Library | Eigen 3.3.4, OpenCV4.4.5 |

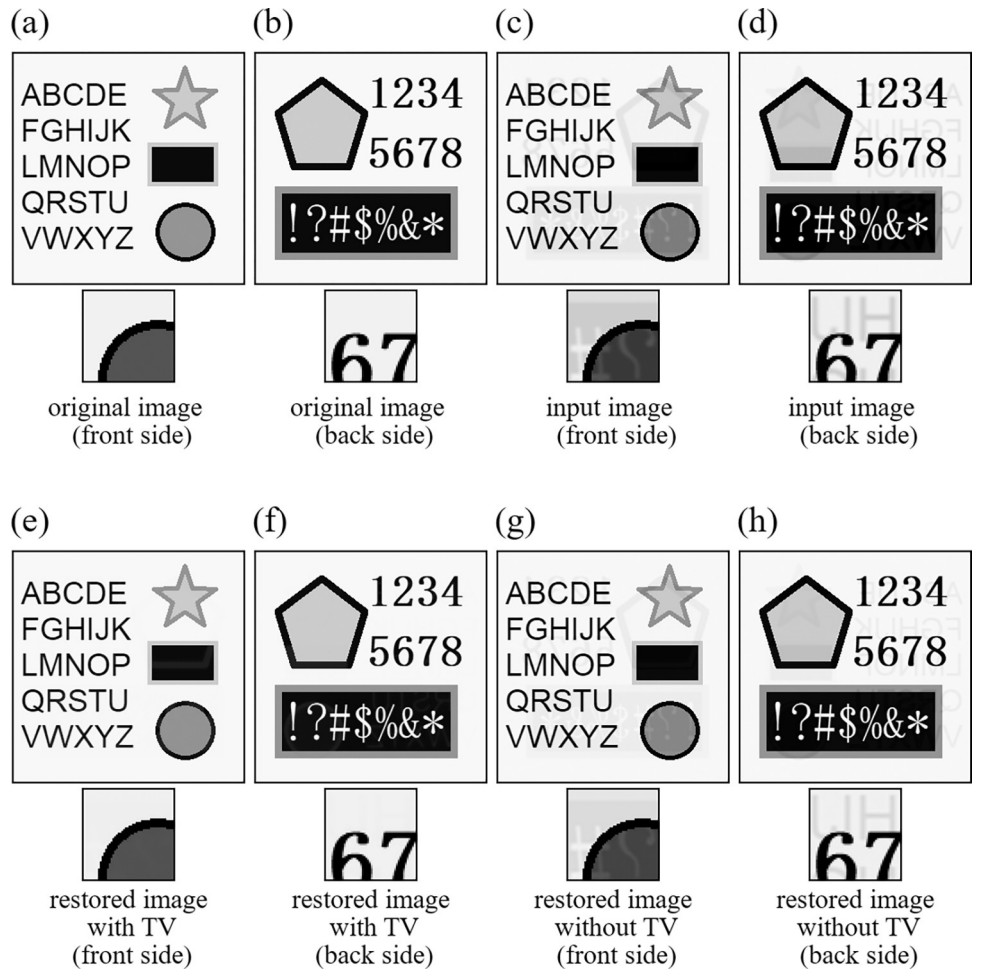

**Fig 3. Images of Experiment 1.** (a) original image (front side), (b) original image (back side), (c) input image (front side), (d) input image (back side), (e) restored image with TV (front side), (f) restored image with TV (back side), (g) restored image without TV (front side), (h) restored image without TV (back side).

matters in the front side and the back side, respectively. Also, spatial resolution of these images was 256×256 (pixels). The degraded images by the show-through are shown in Fig 3(C) and 3 (D) for Experiment 1 (Fig 4(C) and 4(D) for Experiment 2). The only difference between Experiment 1 and Experiment 2 lies in the original images, where there exist no real image parts having gradation in Experiment 1 and some parts consist of a real image in Experiment 2. The restored images are shown in Fig 3(E)–3(H) for Experiment 1 (Fig 4(E)–4(H) for Experiment 2), where a small part of each image is also shown at the bottom after applying gamma-correction to enhance the difference in restoration. The values of evaluation indices (PSNR and SSIM) are summarized in Table 2. In this experiment, when the TV regularization is not used, the effect of show-through still remains in the restored images, but it could be removed with almost no error values with the TV regularization, leading to improving PSNR and SSIM values significantly.

**5.1.2 Effect of regularization parameters on image quality.** The proposed method involves two regularization parameters (*i.e.* $\alpha$ and $\beta$). This section demonstrates the effect of $\alpha$ and $\beta$ on quality of the restored image. The first parameter $\alpha$ in Eq (16) is the parameter to evaluate the magnitude of the nuclear norm. However, the proposed method with the

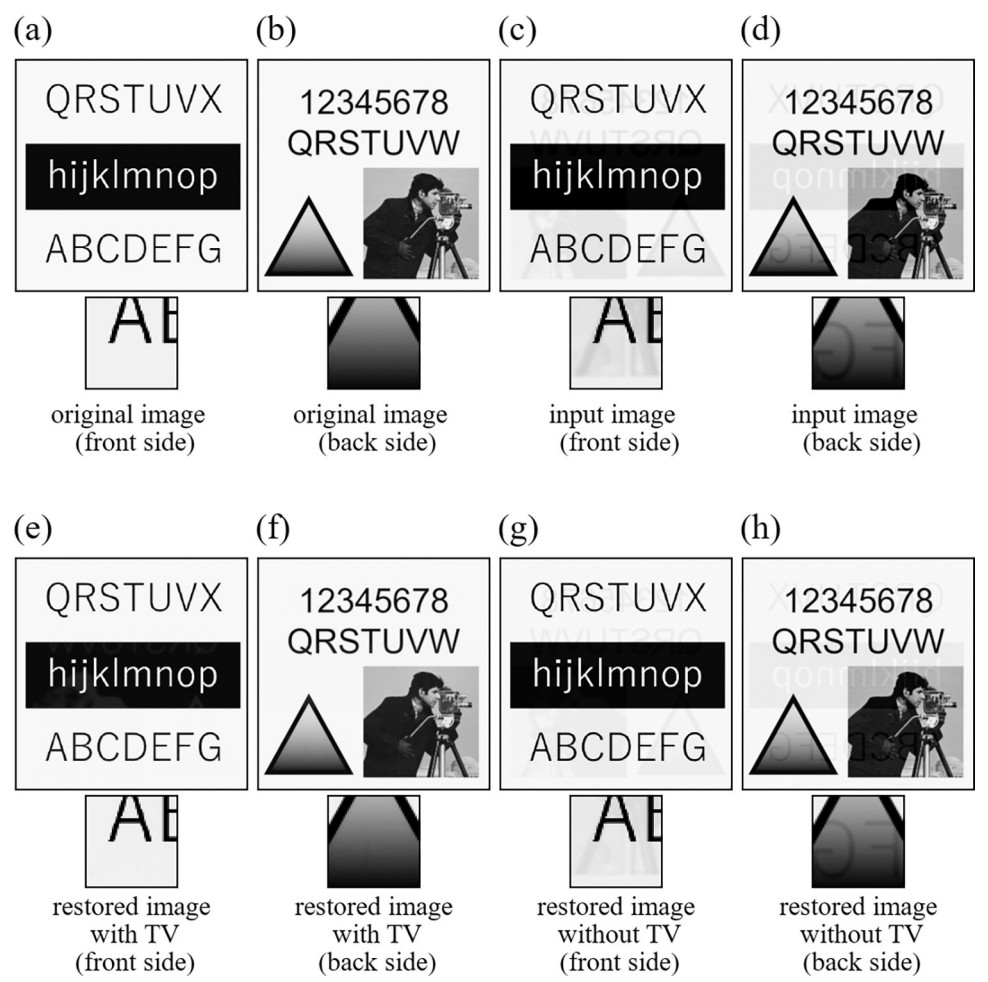

**Fig 4. Images of Experiment 2.** (a) original image (front side), (b) original image (back side), (c) input image (front side), (d) input image (back side), (e) restored image with TV (front side), (f) restored image with TV (back side), (g) restored image without TV (front side), (h) restored image without TV (back side).

**Table 2. Values of evaluation indices (PSNR and SSIM) in Experiment 1 and Experiment 2.**

| | | | PSNR | SSIM |
|---|---|---|---|---|
| **Experiment 1** | Front side | Initial value | 26.953 | 0.941 |
| | | With TV | **39.067** | **0.995** |
| | | Without TV | 31.263 | 0.972 |
| | Back side | Initial value | 30.533 | 0.919 |
| | | With TV | **40.521** | **0.993** |
| | | Without TV | 34.862 | 0.969 |
| **Experiment 2** | Front side | Initial value | 28.086 | 0.905 |
| | | With TV | **39.431** | **0.991** |
| | | Without TV | 32.351 | 0.967 |
| | Back side | Initial value | 26.079 | 0.927 |
| | | With TV | **41.980** | **0.993** |
| | | Without TV | 30.074 | 0.956 |

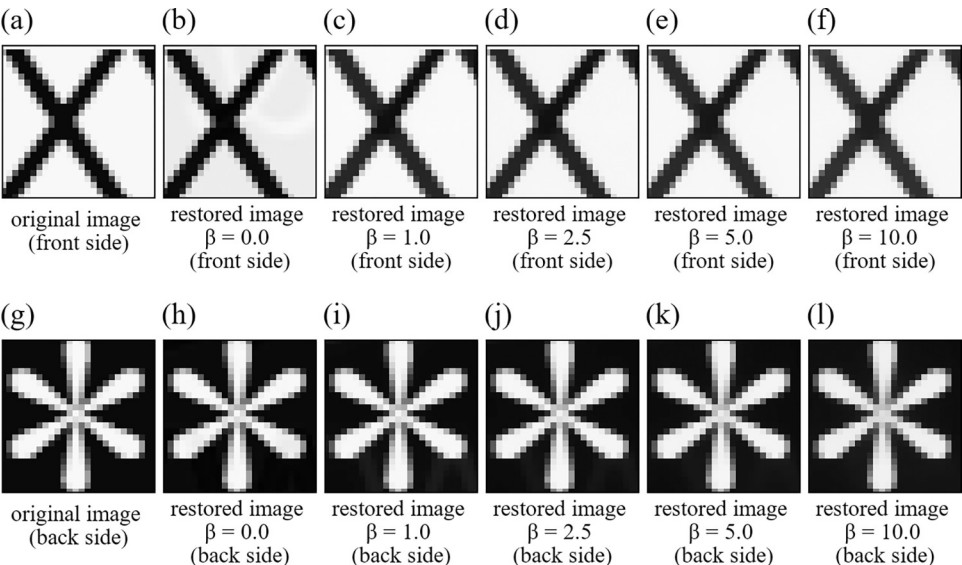

**Fig 5. Zoomed small parts of the restored images in Experiment 3.** (a) original image (front side), (b) restored image $\beta = 0.0$ (front side), (c) restored image $\beta = 1.0$ (front side), (d) restored image $\beta = 2.5$ (front side), (e) restored image $\beta = 5.0$ (front side), (f) restored image $\beta = 10.0$ (front side), (g) original image (back side), (h) restored image $\beta = 0.0$ (front side), (i) restored image $\beta = 1.0$ (back side), (j) restored image $\beta = 2.5$ (back side), (k) restored image $\beta = 5.0$ (back side), (l) restored image $\beta = 10.0$ (back side).

recommended AGP + SVP optimization method does not need to adjust $\alpha$. This is because it uses the singular value projection (SVP) instead of the nuclear norm reduction, which directly imposes the constraint that the rank of $X$ is one by computing only one largest singular value and discarding other singular values. The second parameter $\beta$ affects quality of the restored image. To evaluate the effect of $\beta$ on the image quality, the same simulation experiments as in Experiment 1 and Experiment 2 were performed with five different values of $\beta$ (*i.e.* $\beta = 0.0, 1.0, 2.5, 5.0, 10.0$). These experiments are named as Experiment 3 and Experiment 4, respectively. Figs 5 and 6 show zoomed small parts of the restored images for each value of $\beta$. Table 3 summarizes PSNR and SSIM indices for each value of $\beta$. Fig 7 shows changes of PSNR and SSIM indices with different $\beta$ values. It is observed that increasing $\beta$ value has no significant effect on large edge parts such as those in Figs 5(A), 5(B) and 6(A), but it does smooth out fine edges such as those in Fig 6(B). From Fig 7 and Table 3, it is also observed that if $\beta$ is made too large, PSNR and SSIM indices become worse and $\beta = 2.5$ yields the best restoration accuracy in average for the setup of Experiments 3 and 4. Finally, we mention that the image quality was not very sensitive to $\beta$ value than what we expected before these experiments.

**5.1.3 Experiment to evaluate convergence.** The convergence of the proposed method with the recommended APG + SVP optimization method was evaluated. The same simulation experiments as in Experiment 1 and Experiment 2 were performed with 1000 iterations of APG + SVP method. These experiments are named as Experiment 5 and Experiment 6, respectively. The changes in PSNR and SSIM indices are shown in Fig 8. From Fig 8, it is observed that both PSNR and SSIM converges at 200 iterations and the convergence is stable with no divergence or oscillating behavior. From this experiment, it can be said that the convergence of the method is guaranteed and 200 iterations are sufficient for the convergence.

**5.1.4 Experiment to confirm the effects of APG and SVP.** Next, we confirmed the effects of APG and SVP introduced in Section 4.2.2. Image restorations were performed by changing the combination of APG (or PG) and SVP (or ST) to determine which iterative method

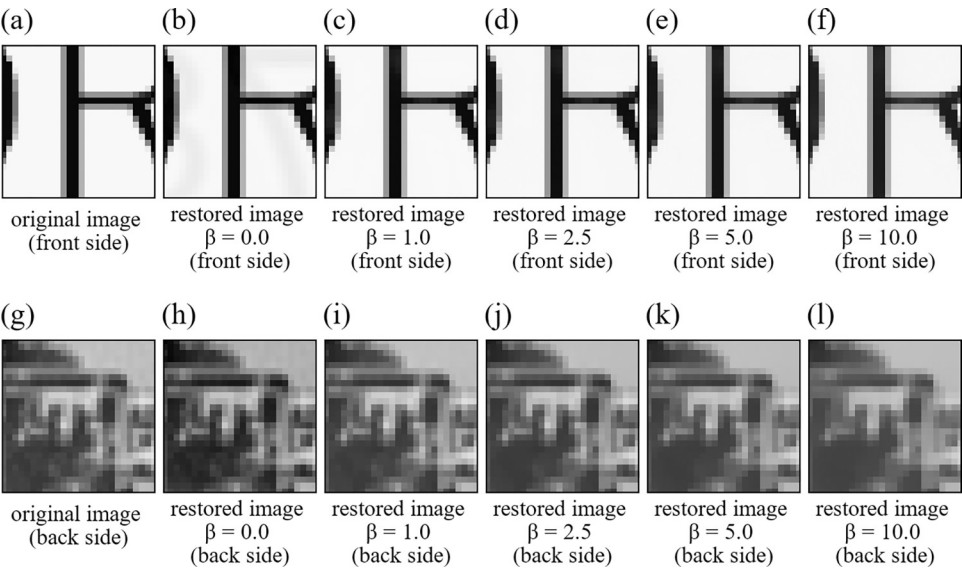

**Fig 6. Zoomed small parts of the restored images in Experiment 4.** (a) original image (front side), (b) restored image $\beta = 0.0$ (front side), (c) restored image $\beta = 1.0$ (front side), (d) restored image $\beta = 2.5$ (front side), (e) restored image $\beta = 5.0$ (front side), (f) restored image $\beta = 10.0$ (front side), (g) original image (back side), (h) restored image $\beta = 0.0$ (front side), (i) restored image $\beta = 1.0$ (back side), (j) restored image $\beta = 2.5$ (back side), (k) restored image $\beta = 5.0$ (back side), (l) restored image $\beta = 10.0$ (back side).

converges faster. The input images in Experiment 7 and Experiment 8 were the same as those of Experiment 1 and Experiment 2, respectively. Image restorations were performed with the following four different combinations: APG + SVP (*i.e.* the method proposed in this paper), PG + SVP, APG + ST, and PG + ST. In this experiment, we judged that the iteration converged

**Table 3. Values of evaluation indices (PSNR and SSIM) in Experiment 3 and Experiment 4.**

|  |  | $\beta$ | PSNR | SSIM |
|---|---|---|---|---|
| **Experiment 3** | **Front side** | 0.0 | 31.263 | 0.972 |
|  |  | 1.0 | 38.582 | 0.994 |
|  |  | 2.5 | **39.067** | **0.995** |
|  |  | 5.0 | 37.314 | 0.994 |
|  |  | 10.0 | 34.822 | 0.993 |
|  | **Back side** | 0.0 | 34.862 | 0.969 |
|  |  | 1.0 | 39.293 | 0.988 |
|  |  | 2.5 | **40.521** | **0.993** |
|  |  | 5.0 | 38.668 | 0.992 |
|  |  | 10.0 | 36.100 | 0.991 |
| **Experiment 4** | **Front side** | 0.0 | 32.351 | 0.967 |
|  |  | 1.0 | 38.456 | 0.986 |
|  |  | 2.5 | **39.431** | **0.991** |
|  |  | 5.0 | 38.090 | **0.991** |
|  |  | 10.0 | 35.420 | 0.990 |
|  | **Back side** | 0.0 | 30.074 | 0.956 |
|  |  | 1.0 | **43.184** | **0.996** |
|  |  | 2.5 | 41.980 | 0.993 |
|  |  | 5.0 | 39.487 | 0.990 |
|  |  | 10.0 | 35.836 | 0.985 |

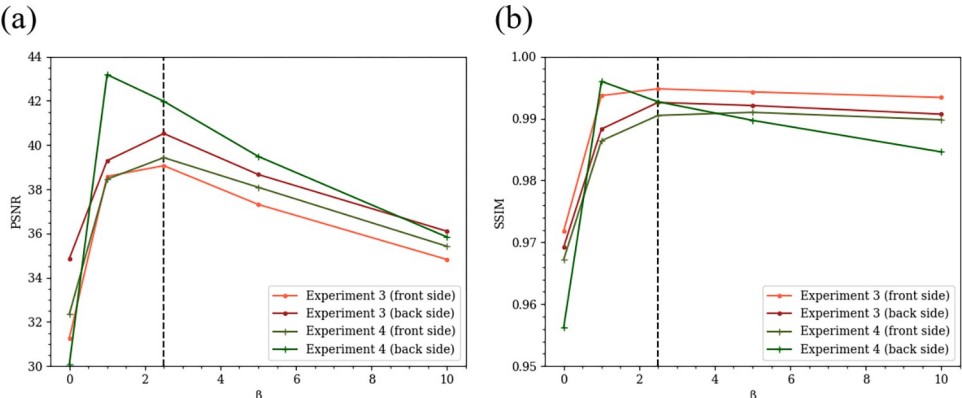

**Fig 7.** Changes of evaluation indices (PSNR and SSIM) according to $\beta$ value in Experiment 3 and Experiment 4. (a) PSNR, (b) SSIM.

when PSNR indices of restored images of both the front and back sides reached to 39 (dB), and the required number of iterations up to the convergence is summarized in Table 4. The changes in PSNR by iterations are shown in Fig 9. As a result, the iteration did not converge with APG + ST and PG + ST. This non-convergence was caused due to the following reason. In APG + ST and PG + ST, the strength of rank 1 constraint imposed on the matrix $X$ is very weak so that a very large number of iterations were necessary up to the convergence. On the other hand, APG + SVP and PG + SVP converged to 39 (dB) in PSNR. However, the number of iterations in APG + SVP was more than 16 times smaller than that of PG + SVP. From this experiment, we conclude that APG + SVP is the best method in terms of convergence speed and computational efficiency.

**5.1.5 Comparison studies with other methods.** To make a comparison with the existing other methods, we chose two typical and well-known methods. The first method is Sharma's method [12], which is based on the adaptive filter. In this method, it is necessary to input a pixel value of the portion without printed matter and a threshold value to determine whether the show-through occurred or not at each pixel. In this experiment, we adjusted values of these parameter through various trial restorations, and used the same values in all experiments. The second method is NMF method [9]. Unlike Sharma's method, this method does not need *a priori* input parameters concerning images to be restored, but the restoration results largely

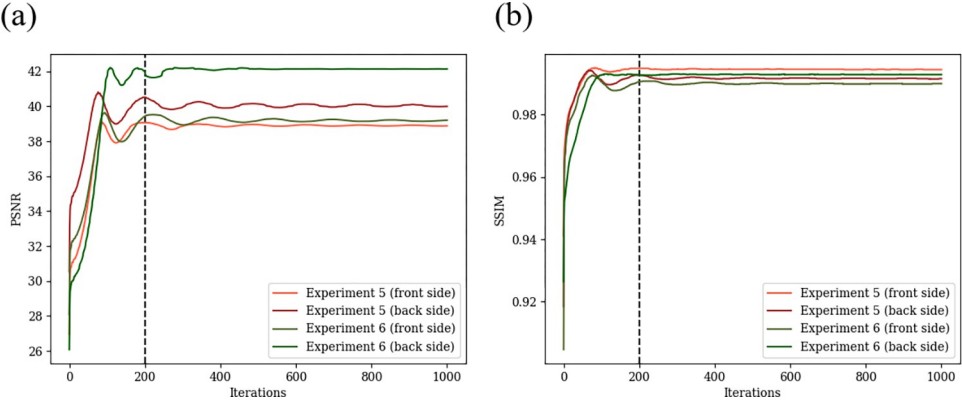

**Fig 8. Changes of evaluation indices (PSNR and SSIM) according to the iteration number in Experiment 5 and Experiment 6.** (a) PSNR, (b) SSIM.

**Table 4. Number of required iterations for each iterative method to reach to 39 (dB) in PSNR for Experiment 7 and Experiment 8.**

| | | NUMBER OF ITERATIONS |
|---|---|---|
| **Experiment 7** | APG + SVP | **85** |
| | PG + SVP | 1543 |
| | APG + ST | - |
| | PG + ST | - |
| **Experiment 8** | APG + SVP | **85** |
| | PG + SVP | 1387 |
| | APG + ST | - |

depend on initial values of the iterative method. In this experiment, we used a random initial value in all experiments. The input images in Experiment 9 and Experiment 10 were the same as those of Experiment 1 and Experiment 2, respectively. All the restored images and the corresponding error images by the three methods are shown in Fig 10 for Experiment 9 (Fig 11 for

(a)

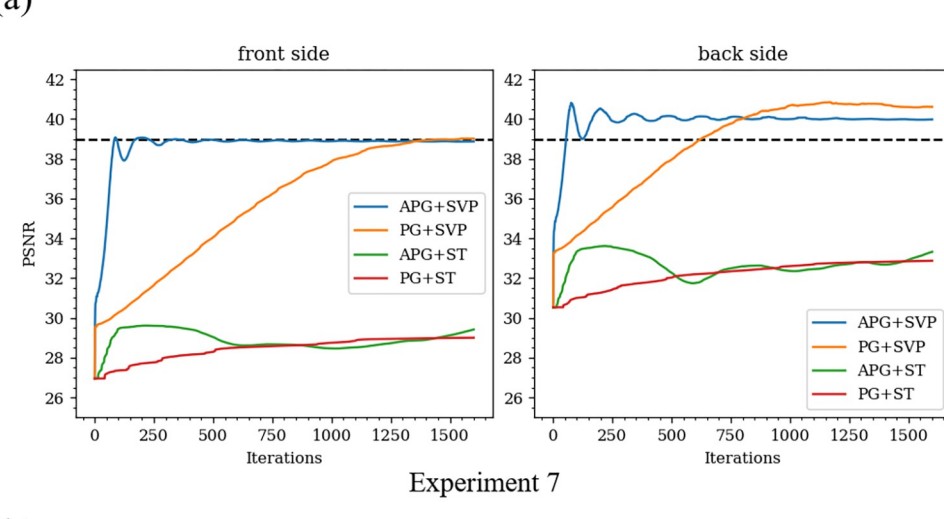

Experiment 7

(b)

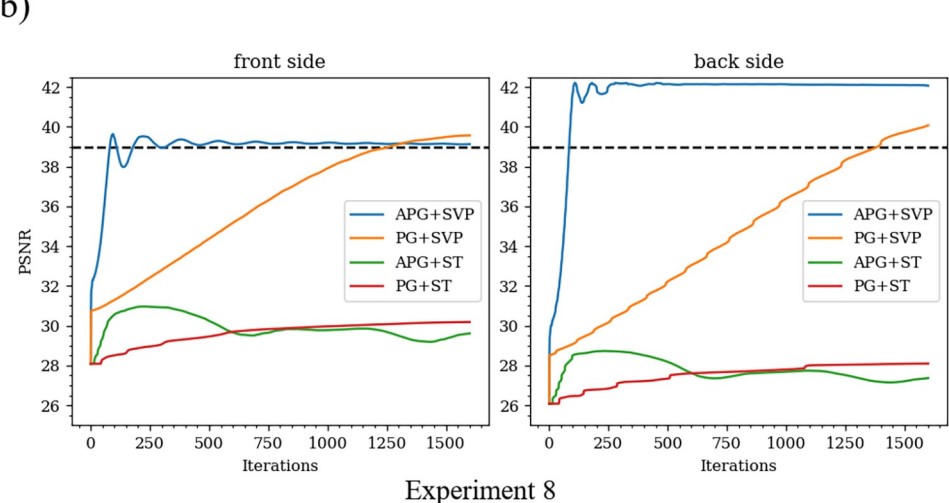

Experiment 8

**Fig 9. Changes of PSNR index according to the iteration number in Experiment 7 and Experiment 8.** (a) Experiment 7, (b) Experiment 8.

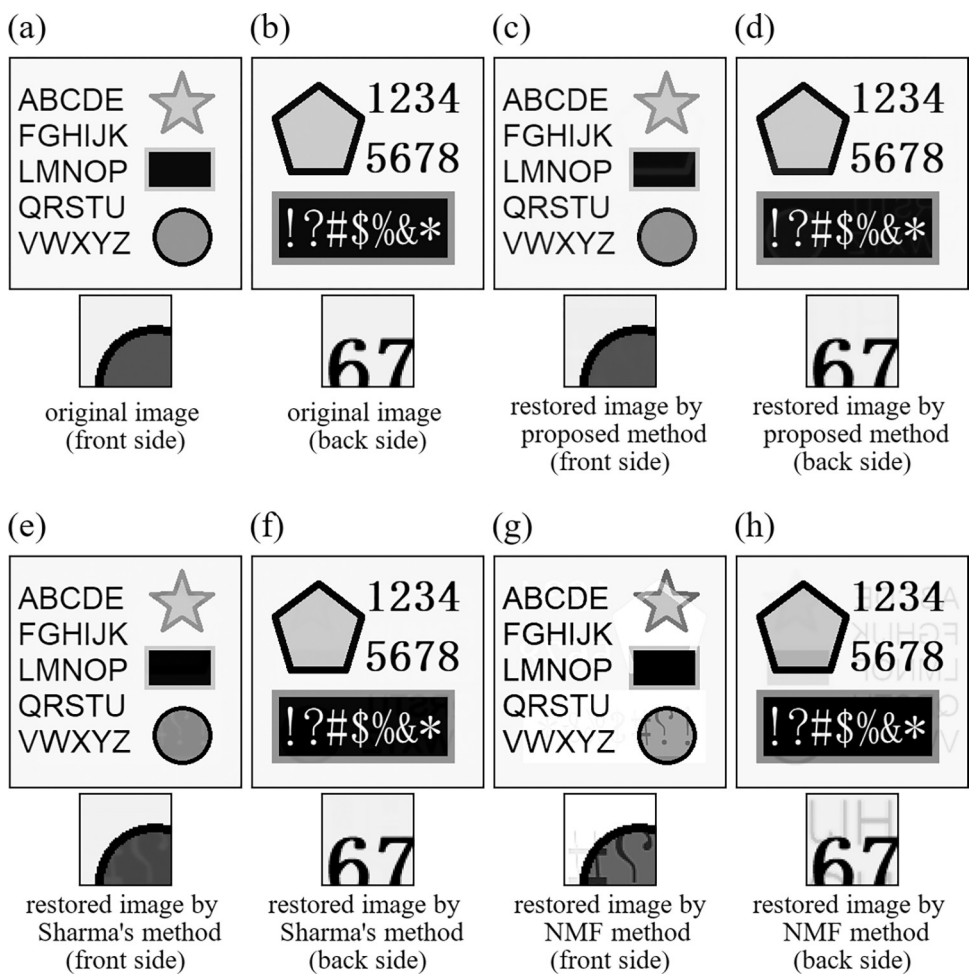

**Fig 10. Images of Experiment 9.** (a) original image (front side), (b) original image (back side), (c) restored image by proposed method (front side), (d) restored image by proposed method (back side), (e) restored image by Sharma's method (front side), (f) restored image by Sharma's method (back side), (g) restored image by NMF method (front side), (h) restored image by NMF method (back side).

Experiment 10). The corresponding values of evaluation indices (PSNR and SSIM) are summarized in Table 5. The show-through still remains in Fig 10(G) and 10(H) in the NMF method, and Figs 10(E) and 11(F) in Sharma's method. However, the proposed method successfully removed the show-through in any image. In Table 5, the proposed method achieved the best PSNR and SSIM values for all the images. From the PSNR and SSIM values, it is clear that the proposed method is superior to the compared other methods.

**5.1.6 Evaluation of computation time.** Computation times of the proposed method and the comparison methods in Experiments 7–10 were evaluated under the environment of Table 1. Table 6 summarizes the computation time for each optimization method in Experiment 7 and Experiment 8. Table 7 summarizes the computation time for each comparison method in Experiment 9 and Experiment 10. From Table 6, it is observed that APG + SVP optimization method reduces the computation time more than 14 times than PG + SVP. APG + ST and PG + ST were not evaluated because they did not converge. From Table 7, it is observed that the proposed method needs a longer computation time than other methods. This drawback needs to be overcome in future work, for example by using parallel processing.

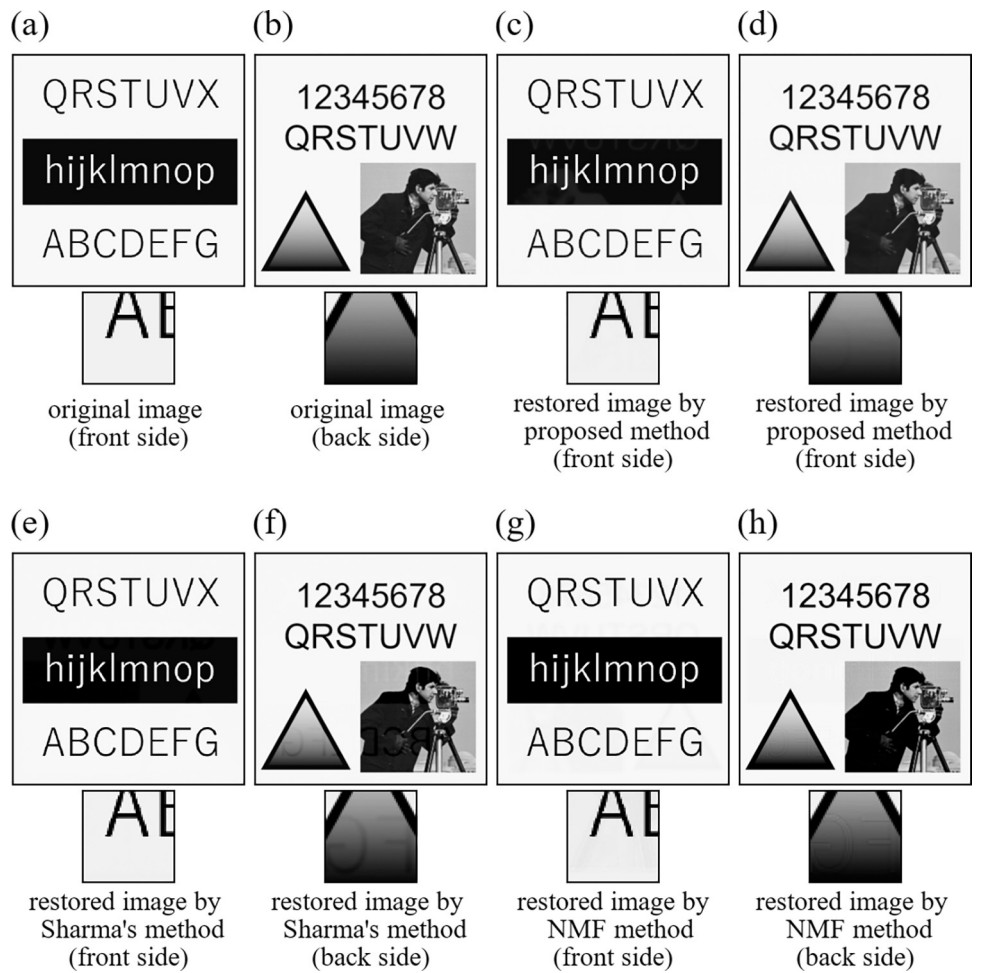

**Fig 11. Images of Experiment 10.** (a) original image (front side), (b) original image (back side), (c) restored image by proposed method (front side), (d) restored image by proposed method (back side), (e) restored image by Sharma's method (front side), (f) restored image by Sharma's method (back side), (g) restored image by NMF method (front side), (h) restored image by NMF method (back side).

## 5.2 Real image experiments

Next, we applied the proposed method, Sharma's method, and the NMF method to images of actual papers scanned by using an electronic scanner. We call these real data experiments as Experiment 11, Experiment 12, and Experiment 13. We used two input images (S2 File) shown in Fig 12(A) and 12(B) for Experiment 11(Fig 13(A) and 13(B) for Experiment 12, Fig 14(A) and 14(B) for Experiment 13). All images used in the real image experiments were acquired by ourselves by scanning real papers having printed materials on the both sides using an electronic scanner. We note that the images used in the three experiments have different characters described below to one another. The images in Experiment 11consist of text parts both in the front and back sides. The images in Experiment 12 consist of text part in the front side and drawing part in the back side. The images in Experiment 13 consist of real images in the front side and text part in the back side. Also, spatial resolution of the input images in Experiment 11, Experiment 12, and Experiment 13 was 256×256 (pixels), respectively. In implementing the proposed method, values of the parameters and initial values of the iteration were set to the same values as in the simulation studies. All the results are summarized in Fig

**Table 5. Values of evaluation indices (PSNR and SSIM) in Experiment 9 and Experiment 10.**

| | | | PSNR | SSIM |
|---|---|---|---|---|
| **Experiment 9** | Front side | Initial value | 26.953 | 0.941 |
| | | Proposed method | **39.067** | **0.995** |
| | | Sharma's method | 36.407 | 0.983 |
| | | NMF method | 26.631 | 0.921 |
| | Back side | Initial value | 30.533 | 0.919 |
| | | Proposed method | **40.521** | **0.993** |
| | | Sharma's method | 37.676 | 0.961 |
| | | NMF method | 28.681 | 0.856 |
| **Experiment 10** | Front side | Initial value | 28.086 | 0.905 |
| | | Proposed method | **39.431** | **0.991** |
| | | Sharma's method | 37.722 | 0.946 |
| | | NMF method | 32.643 | 0.838 |
| | Back side | Initial value | 26.079 | 0.927 |
| | | Proposed method | **41.980** | **0.993** |
| | | Sharma's method | 34.424 | 0.974 |
| | | NMF method | 28.597 | 0.904 |

12 for Experiment 11 (Fig 13 for Experiment 12, Fig 14 for Experiment 13). The NMF method fails to remove the show-trough in Fig 13(H), while it removes even the true printed matter on the front side in Figs 12(G), 14(G), and 14(H). Both Sharma's method and the proposed method could remove the show-through well, but Figs 13(F) and 14(E) shows that Sharma's method also removed the true printed matter in some parts. In the proposed method, such incorrect removals hardly occurred.

## 5.3 Discussion

From the simulation results in this section, the following six facts about the proposed method can be said. First, the show-through removal by the proposed method improves accuracy by using the TV regularization term. Second, the effect of the choice of $\beta$ in Eq (16) affects the accuracy of the restored image. Third, the proposed method has good convergence. Forth, the use of APG + SVP as the iterative solution method is able to speed up convergence, improve computational efficiency, and also improve accuracy. Fifth, the proposed method provides accurate and stable restoration results with the simpler parameter adjustment compared with Sharma's method and the NMF method. The NMF method is a simple method that does not need the parameter adjustment, but it is not practical because initial values of the iteration change the solution significantly. On the other hand, Sharma's method is able to remove the

**Table 6. Computation time for Experiment 7 and Experiment 8.**

| | | COMPUTATION TIME (SEC) |
|---|---|---|
| **Experiment 7** | APG + SVP | **10.20** |
| | PG + SVP | 168.57 |
| | APG + ST | - |
| | PG + ST | - |
| **Experiment 8** | APG + SVP | **10.31** |
| | PG + SVP | 154.37 |
| | APG + ST | - |
| | PG + ST | - |

**Table 7. Computation time for Experiment 9 and Experiment 10.**

|  |  | COMPUTATION TIME (SEC) |
| --- | --- | --- |
| **Experiment 9** | Proposed method | 24.27 |
|  | Sharma's method | **0.35** |
|  | NMF method | 2.60 |
| **Experiment 10** | Proposed method | 24.26 |
|  | Sharma's method | **0.37** |
|  | NMF method | 2.72 |

show-through well if the two parameters, *i.e.* the pixel value of portion without the printed matter and the threshold value to determine whether the show-through occurred or not at each pixel, are set to appropriate values. However, the threshold value needs to be changed according to the density of printed matter, and the pixel value of portion without the printed matter also needs to be changed according to the type of paper. On the other hand, the parameter adjustment in the proposed method is easier allowing an accurate and stable show-

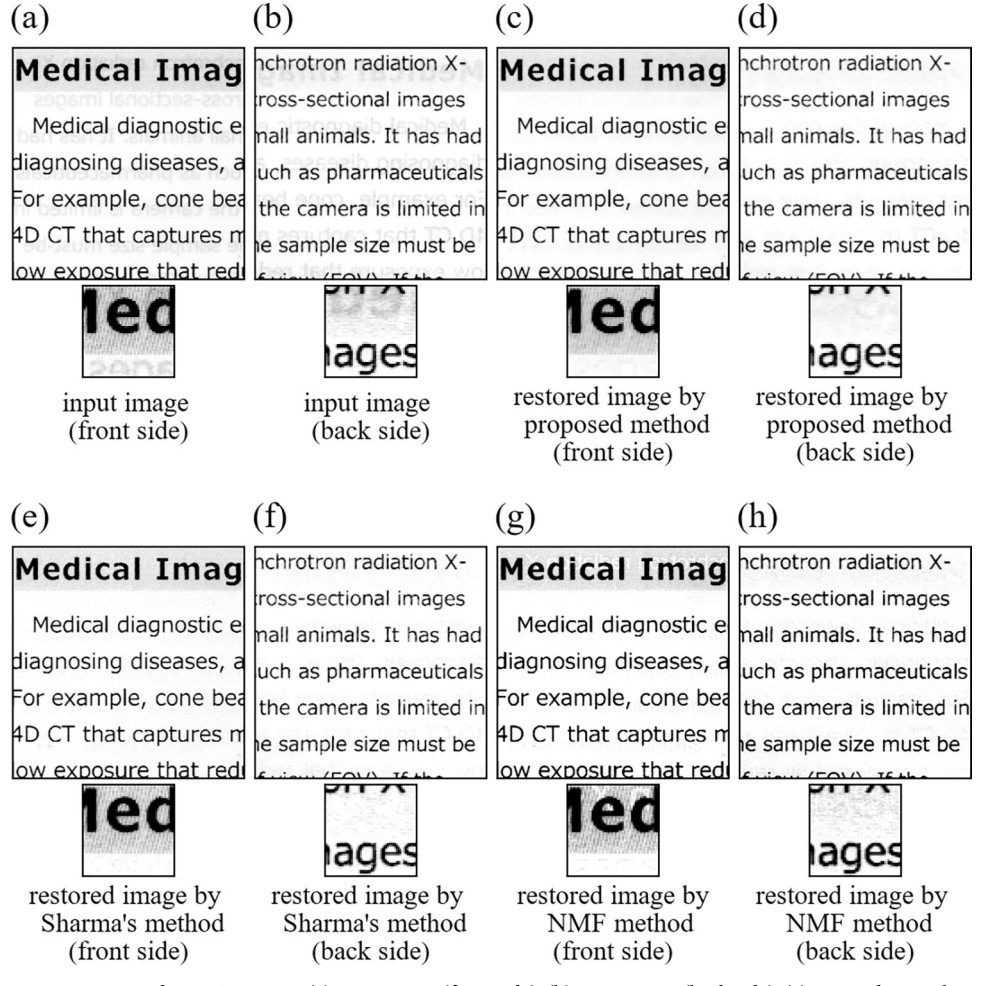

**Fig 12. Images of Experiment 11.** (a) input image (front side), (b) input image (back side), (c) restored image by proposed method (front side), (d) restored image by proposed method (back side), (e) restored image by Sharma's method (front side), (f) restored image by Sharma's method (back side), (g) restored image by NMF method (front side), (h) restored image by NMF method (back side).

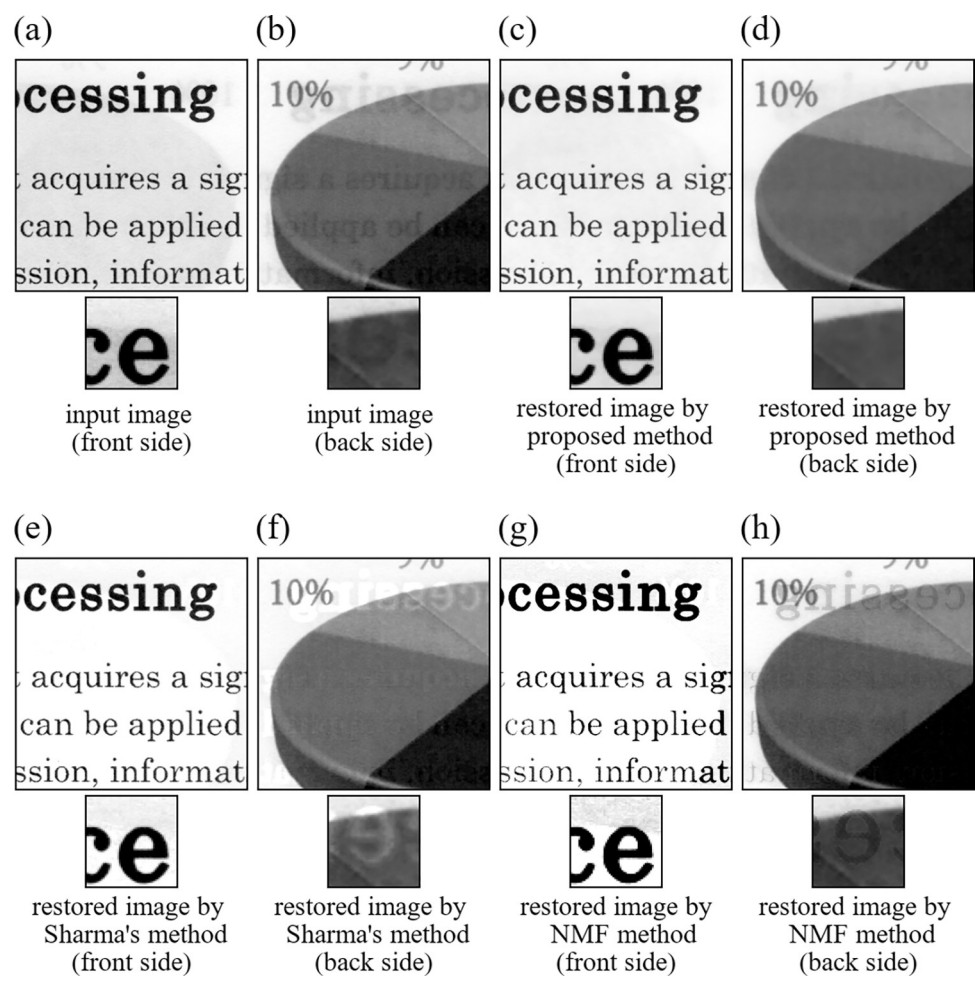

**Fig 13. Images of Experiment 12.** (a) input image (front side), (b) input image (back side), (c) restored image by proposed method (front side), (d) restored image by proposed method (back side), (e) restored image by Sharma's method (front side), (f) restored image by Sharma's method (back side), (g) restored image by NMF method (front side), (h) restored image by NMF method (back side).

through removal for various different images under different conditions. Sixth, the proposed method has a longer computation time than other methods.

## 6 Conclusion

In this paper, we proposed a new method of the show-through removal which takes the blurring into account by applying Ahmed's BID method. From the results of experimental studies, it became clear that the proposed method is able to achieve accurate and stable restorations with the relatively simple parameter adjustments. However, the proposed method still has a limitation that values of the transmittance and the P.S.F. need to be same at every position on the paper. In addition, these values need to be same in the front side and in the back side. In future work, it is necessary to overcome this limitation by developing a space-variant degradation model of show-through and a corresponding restoration method, which better match to real situations. Finally, the most important point to be improved in this method is the reduction of computation time. Reducing computational complexity and increasing convergence speed are also desired.

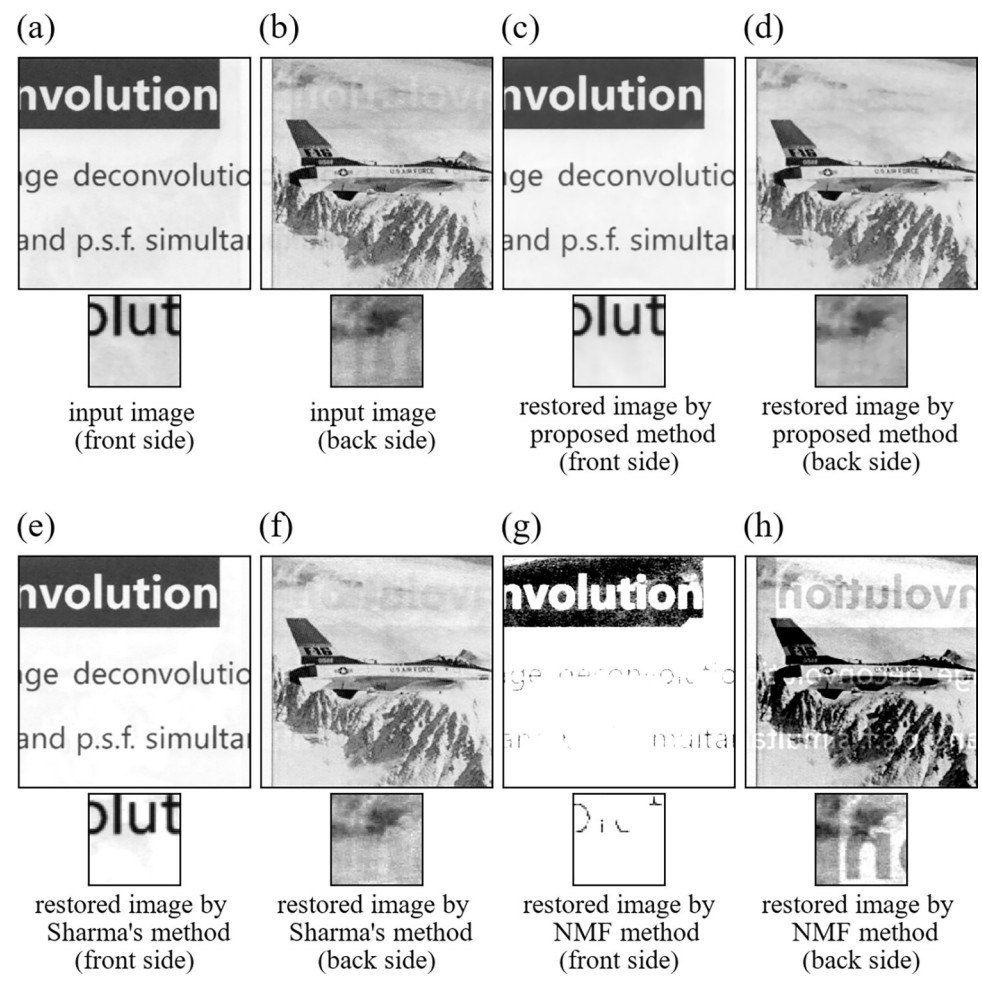

**Fig 14. Images of Experiment 13.** (a) input image (front side), (b) input image (back side), (c) restored image by proposed method (front side), (d) restored image by proposed method (back side), (e) restored image by Sharma's method (front side), (f) restored image by Sharma's method (back side), (g) restored image by NMF method (front side), (h) restored image by NMF method (back side).

## Supporting information

**S1 File. Original images used in the simulation experiments.**
(ZIP)

**S2 File. Images used in the real image experiments.**
(ZIP)

## Acknowledgments

We appreciate Mr. Yoshitaka Izumi and Mr. Dan Suto for useful discussions on this work.

## Author Contributions

**Conceptualization:** Sota Kawakami.

**Data curation:** Sota Kawakami.

**Formal analysis:** Sota Kawakami.

**Investigation:** Sota Kawakami.

**Methodology:** Sota Kawakami.

**Project administration:** Sota Kawakami.

**Resources:** Sota Kawakami.

**Software:** Sota Kawakami.

**Supervision:** Hiroyuki Kudo.

**Validation:** Sota Kawakami.

**Visualization:** Sota Kawakami.

**Writing – original draft:** Sota Kawakami.

**Writing – review & editing:** Hiroyuki Kudo.

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
