## [Decision Letter · Decision Letter 0]

16 Feb 2024

PONE-D-24-02202Show-through removal with sparsity-based blind deconvolutionPLOS ONE

Dear Dr. Kawakami,

Thank you for submitting your manuscript to PLOS ONE. After careful consideration, we feel that it has merit but does not fully meet PLOS ONE’s publication criteria as it currently stands. Therefore, we invite you to submit a revised version of the manuscript that addresses the points raised during the review process.

We look forward to receiving your revised manuscript.

Kind regards,

Abel C.H. Chen

Academic Editor

PLOS ONE

Reviewers' comments:

Reviewer's Responses to Questions

**Comments to the Author**

1. Is the manuscript technically sound, and do the data support the conclusions?

Reviewer #1: Yes

Reviewer #2: Partly

2. Has the statistical analysis been performed appropriately and rigorously? 

Reviewer #1: Yes

Reviewer #2: I Don't Know

3. Have the authors made all data underlying the findings in their manuscript fully available?

Reviewer #1: Yes

Reviewer #2: Yes

4. Is the manuscript presented in an intelligible fashion and written in standard English?

Reviewer #1: Yes

Reviewer #2: Yes

5. Review Comments to the Author

Reviewer #1: In this paper, the blind image deconvolution problem, which is the original image and the point spread function (PSF) are unknown, has been studied. This type of problem cannot be directly solved due to ill-conditioned, and therefore minimization methods are used for this type of problem. The subject under study is a practical one that holds interest for researchers in the field of image processing. While the paper is well-written both scientifically and in terms of its text, it needs to be improved. Some of the corrections that need to be made are listed below.

1- The time required for the algorithm should be examined through an example.

2- To enhance interpretability, the authors should specify the type of system employed for simulating the algorithm. Details such as the simulation environment (Matlab, C, Python, etc.), hardware (CPU or GPU), operating system (Windows or Linux), and other relevant information should be included in the text.

3- The discussion of the convergence of the presented method may be lengthy and challenging in terms of theory. However, it is recommended that the authors demonstrate the convergence of the method in the numerical results section by plotting the changes in PSNR and SSIM values against the number of iterations.

4- In order to be able to simulate the algorithm, the source of the images used in the paper must be stated.

5- The selection of the regularization parameters (\\alpha and \\beta) in the total variation problems are a challenging issue and various methods have been introduced to select these parameters. In the current paper, these parameters have been selected as a fixed number. However, different results are obtained by changing these parameters. The authors are advised to calculate the PSNR and SSIM values in a table for different values of these parameters.

Reviewer #2: Why you didn't use autoencoder as anomaly detection? Autoencoders is a good self-supervised machine learning technique which can use as anomaly detection (Autoencoders and their Applications inMachine Learning: A Survey, DOI: 10.1007/s10462-023-10662-6).

6. PLOS authors have the option to publish the peer review history of their article (what does this mean?). If published, this will include your full peer review and any attached files.

Reviewer #1: No

Reviewer #2: No

---

## [Author Response · Author response to Decision Letter 0]

22 Apr 2024

Thanks to all the reviewers. 

Thank you for your comments and helpful suggestions to improve our paper. We have incorporated all your suggestions into the revised version as explained below.

Response to the reviewer #1

Consern#1: The time required for the algorithm should be examined through an example.

We added the computation times of our simulation experiments in Tables 6 and 7 (Page 17 and 18). Furthermore, Section 5.1.6 was added to explain Tables 6 and 7 (Page 17). From Table 6, it is observed that the optimization method using APG + SVP described in Section 4.2.2 significantly reduces the computation time compared with the PG + SVP. From Table 7, it is observed that the proposed method needs a longer computation time than the other comparison methods. This drawback needs to be overcome in future work, for example by using parallel processing.

Consern#2: To enhance interpretability, the authors should specify the type of system employed for simulating the algorithm. Details such as the simulation environment (Matlab, C, Python, etc.), hardware (CPU or GPU), operating system (Windows or Linux), and other relevant information should be included in the text.

We added Table 1 which summarizes the hardware and software environment used in the experimental studies in Section 5.1 (Page 11). Furthermore, one sentence was added to explain Table 1 (Page 11).

Consern#3: The discussion of the convergence of the presented method may be lengthy and challenging in terms of theory. However, it is recommended that the authors demonstrate the convergence of the method in the numerical results section by plotting the changes in PSNR and SSIM values against the number of iterations.

We performed new experiments (experiments 5 and 6) in Section 5.1.3 to evaluate the convergence of the proposed method with the recommended APG + SVP optimization method. We added Fig 8 which shows graphs of the changes of PSNR and SSIM indices according to the iteration number (Page 14). Furthermore, Section 5.1.3 was added to explain Fig 8 (Page 14). From Fig 8, it is observed that both PSNR and SSIM converges at 200 iterations and the convergence is stable with no divergence or oscillating behavior. From this experiment, we think that it can be said that the convergence of the method is guaranteed and 200 iterations are sufficient for the convergence.

In addition, in the experiments of the optimization methods in Section 5.1.4, we added Fig 9, which shows graphs of the changes of PSNR index for each optimization method (Page 15). Furthermore, a sentence was added to explain Fig 9 (Page 15). It is observed that the recommended APG + SVP method converged fastest among all the four investigated optimization methods.

Consern#4: In order to be able to simulate the algorithm, the source of the images used in the paper must be stated.

All images used in the simulation studies were artificially generated by ourselves. We added a sentence to state this fact in Section 5.1 (Page 11). All images used in the real image experiments were acquired by ourselves by scanning real papers having printed materials on the both sides using an electronic scanner. We added a sentence to state this fact in Section 5.2 (Page 18).

For other researchers, all the image files we used in the experiments are attached to S1 (images used in the simulation experiments) and S2 (images used in the real image experiments) in Supporting Information Section (Page 20).

Consern#5: The selection of the regularization parameters (\\alpha and \\beta) in the total variation problems are a challenging issue and various methods have been introduced to select these parameters. In the current paper, these parameters have been selected as a fixed number. However, different results are obtained by changing these parameters. The authors are advised to calculate the PSNR and SSIM values in a table for different values of these parameters.

The proposed method involves two regularization parameters (i.e. α and β). We added Section 5.1.2 (Page 13), which explains the effect of α and β on quality of the restored image. This section includes results of newly performed experiments (Figs 5, 6, 7 and Table 3) (Page 13 and 14), which demonstrate the effect of β on the image quality. We believe that this answers to your comments in a good way. We repeat the added Section 5.1.2 in the next paragraph for your convenience.

The proposed method involves two regularization parameters (i.e. α and β). This section demonstrates the effect of α and β on quality of the restored image. The first parameter α in Eq ([Disp-formula pone.0305208.e037]) is the parameter to evaluate the magnitude of the nuclear norm. However, the proposed method with the recommended AGP + SVP optimization method does not need to adjust α. This is because it uses the singular value projection (SVP) instead of the nuclear norm reduction, which directly imposes the constraint that the rank of X is one by computing only one largest singular value and discarding other singular values. The second parameter β affects quality of the restored image. To evaluate the effect of β on the image quality, the same simulation experiments as in Experiment 1 and Experiment 2 were performed with five different values of β (i.e. β=0.0,1.0,2.5,5.0,10.0). These experiments are named as Experiment 3 and Experiment 4, respectively. Figs 5 and 6 show zoomed small parts of the restored images for each value of β. Table 3 summarizes PSNR and SSIM indices for each value of β. Fig 7 shows changes of PSNR and SSIM indices with different β values. It is observed that increasing β value has no significant effect on large edge parts such as those in Figs 5(a), (b) and 6(a), but it does smooth out fine edges such as those in Fig 6(b). From Fig 7 and Table 3, it is also observed that if β is made too large, PSNR and SSIM indices become worse and β=2.5 yields the best restoration accuracy in average for the setup of Experiments 3 and 4. Finally, we mention that the image quality was not very sensitive to β value than what we expected before these experiments.

Response to the reviewer #2: Why you didn't use autoencoder as anomaly detection? Autoencoders is a good self-supervised machine learning technique which can use as anomaly detection (Autoencoders and their Applications inMachine Learning: A Survey, DOI: 10.1007/s10462-023-10662-6).

After reading your comments, we performed a literature survey. Consequently, we found several show-through removal methods using Auto-encoder. For example, these include a supervised method (DOI:10.1016/j.patcog.2018.08.011), a self-supervised method (DOI:10.48550/arXiv.2203.04814), and an unsupervised method (DOI:10.1016/j.patcog.2021.108099). All of them are based on binarization. We added the most famous method (DOI:10.1016/j.patcog.2018.08.011) as a reference [14] in Introduction (Page 3). Furthermore, in Introduction, we added our short comment repeated in the next paragraphs on the comparison between the optimization-based approach and the Auto-encoder-based approach (Page 3).

Also, in recent years, a show-through removal method using Convolutional Neural Network (CNN) [13] has also been proposed. Among them, the method based on image binarization using Auto-encoder [14] is promising. The Auto-encoder-based approach like [14] can certainly perform accurate show-through removal with only one-sided input image. However, it has the following drawbacks. Since the method uses the image binarization, images with gradations or shadings cannot be recovered well. Furthermore, their accuracy depends on used training data. On the other hand, our proposed method requires both of an image of the fore side and that of the back side, but can be applied to images with gradations or shadings. At the current stage, we believe that both the optimization-based approach like ours and the Auto-encoder-based approach are meaningful to study, because they have different advantages and disadvantages and the both may be able to be combined in the future to develop more accurate or more convenient methods.

---

## [Decision Letter · Decision Letter 1]

8 May 2024

PONE-D-24-02202R1Show-through removal with sparsity-based blind deconvolutionPLOS ONE

Dear Dr. Kawakami,

Thank you for submitting your manuscript to PLOS ONE. After careful consideration, we feel that it has merit but does not fully meet PLOS ONE’s publication criteria as it currently stands. Therefore, we invite you to submit a revised version of the manuscript that addresses the points raised during the review process.

We look forward to receiving your revised manuscript.

Kind regards,

Abel C.H. Chen

Academic Editor

PLOS ONE

Journal Requirements:

Reviewers' comments:

Reviewer's Responses to Questions

**Comments to the Author**

1. If the authors have adequately addressed your comments raised in a previous round of review and you feel that this manuscript is now acceptable for publication, you may indicate that here to bypass the “Comments to the Author” section, enter your conflict of interest statement in the “Confidential to Editor” section, and submit your "Accept" recommendation.

Reviewer #1: All comments have been addressed

Reviewer #2: All comments have been addressed

2. Is the manuscript technically sound, and do the data support the conclusions?

Reviewer #1: Yes

Reviewer #2: Yes

3. Has the statistical analysis been performed appropriately and rigorously? 

Reviewer #1: Yes

Reviewer #2: Yes

4. Have the authors made all data underlying the findings in their manuscript fully available?

Reviewer #1: Yes

Reviewer #2: No

5. Is the manuscript presented in an intelligible fashion and written in standard English?

Reviewer #1: Yes

Reviewer #2: Yes

6. Review Comments to the Author

Reviewer #1: After reviewing the original text, the revised versions, and the opinions of the referees, as well as considering the answers provided and evaluating the article from a scientific standpoint, I believe that the manuscript in its current form may be considered for the next phase of the editorial process.

Also, as a recommendation for authors, it is suggested that the images and codes used for this article be placed on public repositories like GitHub and its address is mentioned in the text of the article. Additionally, the open-sourcing of the code not only facilitates the dissemination of the paper but also boosts its citation count.

Reviewer #2: All previously provided feedback has been duly addressed. However, it's recommended to consider referencing the most recent work on autoencoders, such as "Autoencoders and their applications in machine learning: a survey (2024)," instead of "A selectional auto-encoder approach for document image binarization (2019)," to ensure the inclusion of up-to-date and comprehensive literature in the manuscript.

7. PLOS authors have the option to publish the peer review history of their article (what does this mean?). If published, this will include your full peer review and any attached files.

Reviewer #1: No

Reviewer #2: No

---

## [Author Response · Author response to Decision Letter 1]

13 May 2024

Thanks to all the reviewers. 

Thank you for your second comments and helpful suggestions to improve our paper. We have incorporated all your suggestions into the revised version as explained below.

Reviewer #1: After reviewing the original text, the revised versions, and the opinions of the referees, as well as considering the answers provided and evaluating the article from a scientific standpoint, I believe that the manuscript in its current form may be considered for the next phase of the editorial process. Also, as a recommendation for authors, it is suggested that the images and codes used for this article be placed on public repositories like GitHub and its address is mentioned in the text of the article. Additionally, the open-sourcing of the code not only facilitates the dissemination of the paper but also boosts its citation count.

Thank you for your comment. According to your advice, I have added the link to repository to get images and codes used in the experiment in section 5.1 (Page 12).

Reviewer #2: All previously provided feedback has been duly addressed. However, it's recommended to consider referencing the most recent work on autoencoders, such as "Autoencoders and their applications in machine learning: a survey (2024)," instead of "A selectional auto-encoder approach for document image binarization (2019)," to ensure the inclusion of up-to-date and comprehensive literature in the manuscript.

Thank you for your comment. As you pointed, our reference to the approach using Auto-encoder was not good in the following sense. We cited a paper on the binarization of document image using Auto-encoder, but this is not a standard usage of Auto-encoder. Since a major application of Auto-encoder is to remove disturbance from images, which has been already proposed in image denoising, speckle reduction in SAR or ultrasound images, and clutter rejection in radar images. The same method can be also used to remove show-through from scanned images. Since the proposed optimization-based method and the Auto-encoder method have different advantages and disadvantages, it would be interesting to compare the two methods in terms of various points of views. So, according to your suggestion, we have made the following changes in the 2-nd revised version. First, we changed the cited paper (Ref. 14) from the previous one to the following new one, which you have introduced to us.

Berahmand, K., Daneshfar, F., Salehi, E.S. et al. Autoencoders and their applications in machine learning: a survey. Artif Intell Rev 57, 28 (2024). https://doi.org/10.1007/s10462-023-10662-6

Second, we have changed the explanation on the use of Auto-encoder (on Page 3) as summarized in the next paragraph.

Also, in recent years, show-through removal methods using Convolutional Neural Network (CNN) have been proposed [13,14]. Among them, the method based on Auto-encoder [14] has a large potential. A typical major application of Auto-encoder is to remove disturbance from signals or images, which has been already proposed in image denoising, speckle reduction in SAR or ultrasound images, and clutter rejection in radar images. The same method can be also used to remove show-through from scanned images. This method requires only a single image of front side. However, its accuracy depends on used training data, and it may be weaker to the show-through having strong correlation with the front page image. On the other hand, our proposed method requires both of an image of the fore side and that of the back side, but using the two images eliminates the necessity of training data and may lead to an improved accuracy (for example, in the above mentioned case). At the current stage, we believe that both the optimization-based approach like ours and the Auto-encoder-based approach are meaningful to study, because they have different advantages and disadvantages and the both may be able to be combined in the future to develop more accurate or more convenient methods.

---

## [Decision Letter · Decision Letter 2]

27 May 2024

Show-through removal with sparsity-based blind deconvolution

PONE-D-24-02202R2

Dear Dr. Kawakami,

We’re pleased to inform you that your manuscript has been judged scientifically suitable for publication and will be formally accepted for publication once it meets all outstanding technical requirements.

Kind regards,

Abel C.H. Chen

Academic Editor

PLOS ONE

Additional Editor Comments (optional):

Reviewers' comments:

Reviewer's Responses to Questions

**Comments to the Author**

1. If the authors have adequately addressed your comments raised in a previous round of review and you feel that this manuscript is now acceptable for publication, you may indicate that here to bypass the “Comments to the Author” section, enter your conflict of interest statement in the “Confidential to Editor” section, and submit your "Accept" recommendation.

Reviewer #1: All comments have been addressed

Reviewer #2: (No Response)

2. Is the manuscript technically sound, and do the data support the conclusions?

Reviewer #1: Yes

Reviewer #2: (No Response)

3. Has the statistical analysis been performed appropriately and rigorously? 

Reviewer #1: Yes

Reviewer #2: (No Response)

4. Have the authors made all data underlying the findings in their manuscript fully available?

Reviewer #1: Yes

Reviewer #2: (No Response)

5. Is the manuscript presented in an intelligible fashion and written in standard English?

Reviewer #1: Yes

Reviewer #2: (No Response)

6. Review Comments to the Author

Reviewer #1: The authors have addressed all the recommendations of the reviewers in a reasonable manner, the manuscript in the current form may be considered for the further phase of the editorial process.

Reviewer #2: The new version of the manuscript correctly do the previous comment.

All the previous comments have been investigated.

7. PLOS authors have the option to publish the peer review history of their article (what does this mean?). If published, this will include your full peer review and any attached files.

Reviewer #1: No

Reviewer #2: **Yes: **fatemeh daneshfar

---

## [Editor Report · Acceptance letter]

3 Jun 2024

PONE-D-24-02202R2 

PLOS ONE

Dear Dr. Kawakami, 

I'm pleased to inform you that your manuscript has been deemed suitable for publication in PLOS ONE. Congratulations! Your manuscript is now being handed over to our production team.

Kind regards, 

on behalf of

Dr. Abel C.H. Chen 

Academic Editor

PLOS ONE